# Functional classification and validation of yeast prenylation motifs using machine learning and genetic reporters

**Brittany M. Berger**[1], **Wayland Yeung**[1,2], **Arnav Goyal**[1], **Zhongliang Zhou**[3],
**Emily R. Hildebrandt**[1], **Natarajan Kannan**[1,2], **Walter K. Schmidt**[1]*

**1** Department of Biochemistry and Molecular Biology, University of Georgia, Athens, Georgia, United States of America, **2** Institute of Bioinformatics, University of Georgia, Athens, Georgia, United States of America, **3** Department of Computer Science, University of Georgia, Athens, Georgia, United States of America

* wschmidt@uga.edu

**Data Availability Statement:** All relevant data are within the paper and its Supporting information files.

**Funding:** This work was supported by NIH funds to WKS and NK (NIH NIGMS GM132606, https://

## Abstract

Protein prenylation by farnesyltransferase (FTase) is often described as the targeting of a cysteine-containing motif (CaaX) that is enriched for aliphatic amino acids at the $a_1$ and $a_2$ positions, while quite flexible at the X position. Prenylation prediction methods often rely on these features despite emerging evidence that FTase has broader target specificity than previously considered. Using a machine learning approach and training sets based on canonical (prenylated, proteolyzed, and carboxymethylated) and recently identified shunted motifs (prenylation only), this study aims to improve prenylation predictions with the goal of determining the full scope of prenylation potential among the 8000 possible Cxxx sequence combinations. Further, this study aims to subdivide the prenylated sequences as either shunted (i.e., uncleaved) or cleaved (i.e., canonical). Predictions were determined for *Saccharomyces cerevisiae* FTase and compared to results derived using currently available prenylation prediction methods. *In silico* predictions were further evaluated using *in vivo* methods coupled to two yeast reporters, the yeast mating pheromone **a**-factor and Hsp40 Ydj1p, that represent proteins with canonical and shunted CaaX motifs, respectively. Our machine learning-based approach expands the repertoire of predicted FTase targets and provides a framework for functional classification.

## Introduction

CaaX-type protein prenylation refers to the covalent linkage of a farnesyl or geranylgeranyl isoprenoid group (C15 and C20, respectively) to proteins containing a COOH-terminal CaaX motif, where C is an invariant cysteine, $a_1$ and $a_2$ are typically aliphatic residues, and X is one of many amino acids [1]. Farnesyltransferase (FTase) and geranylgeranyltransferase-I (GGTase-I) facilitate the isoprenoid addition to the CaaX cysteine thiol, with GGTase-I targeting the subset of CaaX sequences having Leu, Phe or Met at the X position [2–4]. For many CaaX proteins, initial isoprenylation is followed by proteolysis that removes the aaX tripeptide, mediated by Rce1p or Ste24p, and carboxymethylation of the isoprenylated cysteine, mediated

www.nih.gov/) and funds to WKS (NIH NIGMS R01GM117148, https://www.nih.gov/). The funders had no role in study design, data collection and analysis, decision to publish, or preparation of the manuscript.

**Competing interests:** The authors have declared no competing interests exist.

**Abbreviations:** FPB, FlexPepBind; FTase, farnesyltransferase; GBDT, Gradient Boosting Decision Tree; GGTase-I, geranylgeranyltransferase-I; PrePS, Prenylation Prediction Suite; SGD, *Saccharomyces* Genome Database; SVM, support vector machine.

by isoprenylcysteine carboxyl methyltransferase (ICMT; Ste14p in yeast) [5]. These modifications increase the overall COOH-terminal hydrophobicity of modified proteins and often occur to CaaX proteins well-known to be membrane associated (e.g., Ras GTPases) (Fig 1A).

Despite FTase arguably being the most well characterized enzyme in the CaaX modification pathway, its specificity still remains unclear. Early primary sequence comparisons of known FTase targets often outlined the standard, aliphatic-enriched consensus motif termed CaaX. One of the first methods to predict FTase substrates was developed into the Prenylation Prediction Suite (PrePS) [6]. This method evaluated the last 15 amino acids of known prenylated targets, including many Ras and Ras-related GTPases and a few non-canonical sequences for which evidence of prenylation was previously established, to determine a consensus of physio-biochemical properties important for prenylation, which was then used to predict prenylation.

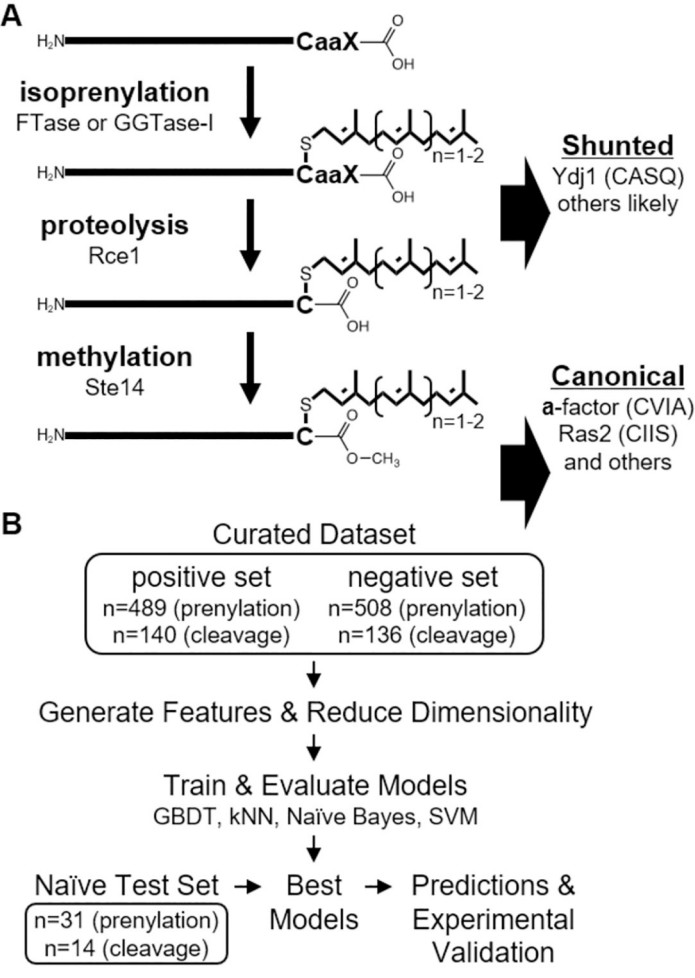

**Fig 1. Biochemical and machine learning workflow diagrams.** A) Modifications occurring to CAAX proteins. Isoprenylation involves attachment of a farnesyl (C15) or geranylgeranyl (C20) lipid to the consensus cysteine amino acid (C) of a COOH-terminal CaaX motif. Shunted CaaX proteins are not further modified. Canonical CaaX proteins undergo proteolytic cleavage to remove the 'aaX' portion of the motif and carboxylmethylation of the isoprenylated cysteine. Examples shown are yeast proteins, but examples exist in other systems. a–aliphatic amino acid; X–one of several amino acids. B) Positive and negative training sets for prenylation and cleavage predictions were curated from published data, used to generate features, then used to train four machine learning algorithms. The trained models were subject to 10-fold cross validation to determine accuracy, precision, recall, and F1-score. The best models, along with a PSSM-based model, were then used to predict prenylation and cleavage outcomes for naïve test sequences that were compared against the experimental observed properties of these sequences.

PrePS was then applied to create a database of all prenylation predictions across all known proteins, regardless of species [7]. The prenylation potential of nearly all 8000 possible CaaX sequences has also been investigated using genetics and high throughput NextGen Sequencing (NGS) in the context of a mutated form of H-Ras (Ras61) that was heterologously expressed in yeast [8]. The identified target sequences were consistent with the initially described consensus CaaX motif. Parallel *in vitro* and *in silico* studies have suggested, however, that FTase may be able to accommodate substantially broader substrates than initially proposed [9–13]. A broader consensus for human FTase was also proposed using FlexPepBind (FPB), an approach involving structure-based molecular docking and energy minimization constraints [12]. This approach identified several sequences that were not initially expected to be prenylated but subsequently biochemically validated as FTase targets. Despite these new experimental observations and advancements in prenylation prediction methods, many prenylated sequences still fail to be accurately predicted as FTase substrates. Past approaches involving *in vitro* peptide libraries and metabolic labeling with farnesyl analogs suitable for click-chemistry have been able to identify additional non-canonical sequences as FTase targets, however, peptide libraries are often costly and can be labor intensive and metabolic labeling is limited to cell specific sequences [9, 10, 14–17]. Thus, limitations still prevent exploration of the full scope of prenylation for all 8000 Cxxx sequences.

While the specificity of FTase is emerging to be more flexible than anticipated, the CaaX proteases that mediate subsequent cleavage of the aaX tripeptide appear more stringent, requiring aliphatic residues at $a_1$ and/or $a_2$ positions [18]. This observation identifies an inherent bias in many FTase assays due to the use of canonical reporters such as Ras and **a**-factor where the specificity of the downstream proteases may limit the prenylatable sequences that can be identified. To overcome this bias, we recently developed *S. cerevisiae* Hsp40 Ydj1p into a novel *in vivo* reporter for yeast FTase activity [19]. Unlike canonical reporters previously used *in vivo*, the non-canonical CaaX sequence of Ydj1p (CASQ) is farnesylated, then "shunted" out of the canonical CaaX pathway without being further proteolyzed and carboxymethylated. Previous studies have shown that yeast require Ydj1p prenylation for proper interactions with Hsp90 and growth at high temperatures, as evident by a growth defect at higher temperatures when canonical modification occurs (i.e., prenylation, proteolysis and carboxymethylation), and a further reduction in growth with lack of prenylation [19–21]. This growth phenotype was used to identify 153 sequences that supported Ydj1p prenylation-dependent yeast growth at high temperatures [22]. The recovered sequences were vastly different than standard canonical CaaX sequences, lacking characteristic aliphatic amino acids but consistent with specificities observed through *in vitro* and *in silico* studies. For clarity, all 8000 sequences are referred to as Cxxx sequences in this study, while predicted prenylated sequences are referred to as CaaX motifs with qualifiers added to specify those that are canonically modified (i.e., cleaved) or shunted (i.e., uncleaved).

In this study, we used machine learning and yeast genetic data derived from both Ras61 and Ydj1p *in vivo* reporters to develop methods for predicting the prenylation potential of all 8000 Cxxx sequences within the yeast system. Predictions were then compared to those derived using PrePS, FPB, and Freq. The latter is a frequency-based, in-house method developed in our previous study of Cxxx sequences that support Ydj1p-dependent thermotolerance. Our findings suggest that the use of machine learning with data derived from both canonical and non-canonical reporters results in improved prediction of yeast FTase targets. This approach was also used to develop a first-ever prediction for CaaX proteolysis, leading to effective predictions for establishing whether a prenylated sequence follows the canonical or shunted pathway (i.e., cleaved vs. uncleaved).

## Materials and methods

### Training set curation

**Prenylation.** Training sets can be found in S1 File and were derived from previously published datasets. The positive set initially included 369 sequences identified through a Ras61 prenylation screen (enrichment score >3 at 37˚C; ≥5 occurrences) and 153 sequences identified through a Ydj1p prenylation screen [8, 22]. The positive training set was curated to form a reduced set of 489 unique sequences by removing duplicate sequences that overlapped between the sets (n = 8), sequences found naturally in the *Saccharomyces cerevisiae* proteome (n = 21), and sequences that had previously been incorporated into reporters (n = 4). The negative set initially consisted of 514 sequences that were lowest scoring in the Ras61 prenylation screen (enrichment score ≤0.036 at 37˚C; ≥5 occurrences at 25˚C). The negative set was curated to form a reduced set of 508 unique sequences by removing 6 sequences found naturally in the *Saccharomyces cerevisiae* proteome.

**Cleavage.** Training sets can be found in S1 File and were derived from previously published datasets [8, 22]. The positive set initially included 153 top scoring Ras61 sequences (enrichment score >3 at 37˚C; ≥5 occurrences). From this, the positive training set was reduced to a unique set of 140 by removing duplicate sequences that overlapped with the Ydj1p set (n = 2), sequences found naturally in the *Saccharomyces cerevisiae* proteome (n = 8), and sequences that had previously been incorporated into reporters (n = 3). The negative set initially included 153 sequences recovered in theYdj1p screen. The negative set was reduced to 136 sequences by removing sequences that were genetically confirmed to be canonically modified (n = 15), sequences found naturally in the *Saccharomyces cerevisiae* proteome (n = 1), and sequences that had previously been incorporated into reporters (n = 1).

### Feature generation & pre-processing

**Feature generation.** In order to generate features for machine learning, we explored three different ways of representing Cxxx sequences: 1) the specific amino acid sequence represented by one-hot encoding, 2) the physico-biochemical features retrieved from the AAindex database (ftp://ftp.genome.jp/pub/db/community/aaindex/; downloaded 1/17/2021) [23], and 3) sequence embedding generated by ESM-1b (https://github.com/facebookresearch/esm; downloaded 2/9/2021), a state-of-the-art Transformer model that was pre-trained on roughly 250M protein sequences [24]. Sequence features were represented by an array of size 60, which accounts for one-hot encoding of 20 amino acid residues at the 3 variable "x" positions of the Cxxx sequence. AAindex features were represented by an array of size 1659, which accounts for all 553 physico-biochemical features defined by the database for each of the 3 positions. These features were normalized to a range of 0 to 1 in order to equalize their scales. ESM-1b features were generated by taking advantage of the model's ability to account for contextual information, capturing the potential effects of neighboring residues. We represented the COOH-terminal localization of the Cxxx sequence by front-padding with 100 unspecified "x" residues. In addition, the model added two special characters to represent the beginning and end of the amino acid sequence. This sequence was used to generate an embedding of size (1280, 106), which represents a 1280-dimensional abstract description of 104 residue positions plus two special symbols. ESM-1b features were extracted from this embedding by retrieving the positions corresponding with the Cxxx sequence and end-of-sequence character, which resulted in an array of size (1280, 5), flattened to size 6400. We retained the positional encoding corresponding to the invariant cysteine due to the model's unique ability to capture contextual information.

*Dimensionality reduction.* Redundant features were removed through principal component analysis, a standard dimensionality reduction technique [25]. This resulted in the reduction of sequence features from 60 to 53 dimensions, AAindex features from 1659 to 50 dimensions, and ESM-1b features from 6400 to 276 dimensions. These reduced features captured 99% of total variance in each feature set.

## Prediction of Cxxx prenylation & cleavage

**Scoring.** We quantified the performance of all prediction models based on accuracy, precision, recall, and F1-score. Reported values indicate the mean across 10-fold cross validation while confidence intervals indicate the standard deviation.

**Position-specific scoring matrix (PSSM).** We constructed a PSSM based on prenylated or cleaved motifs. The amino acid distribution was normalized against a background amino acid distribution defined by the BLOSUM62 substitution matrix [26] with a pseudo-count of 0.05. The resulting model was used to calculate the log probability of a given sequence being prenylated or cleaved. In order to obtain binary predictions, we defined a cutoff log probability that best separated the positive from the negative examples.

**Machine learning algorithms.** We tested the performance of various machine learning algorithms as implemented by Scikit-learn [27]. The parameters of individual predictors were optimized by grid search. Specific algorithms tested were support vector machine (SVM), Naïve Bayes, k-nearest neighbor (kNN), and Gradient Boosted Decision Tree (GBDT). In subsequent analyses, we estimated the probabilities of each prediction for SVM through Platt scaling [28].

**Software.** All computational analyses, unless otherwise mentioned, were implemented in Python 3 using NumPy [29] and PyTorch [30]. Figure plots were created using Matplotlib [31], seaborn [32], WebLogo3 [33], and Adobe Illustrator. For WebLogo3, a custom color scheme was used where cysteine (C) was blue, polar charged amino acids (H, K, R, E, D) were green, polar uncharged amino acids (N, Q, S, T, Y) were black, branched-chain amino acids (L, I, V) were red, and all other amino acids (F, A, P, G, M, W) were purple. This scheme matches that used in a previously published study of FTase specificity by our group [22].

**Cut-offs used for predictions by prenylation methods.** For analysis with the Prenylation Prediction Suite (PrePS; https://mendel.imp.ac.at/PrePS), all 8000 Cxxx sequences were evaluated in the context of human H-Ras (RQHKLRKLNPPDESGPGCMSCKCxxx). While PrePS only requires 15 amino acids for scoring, 26 were used to remain consistent with previous studies [19, 22]. For PrePS, sequences scoring greater than -2 were deemed positive predictions. For FlexPepBind, sequences scoring greater than -1.1 were deemed positive predictions, consistent with the stringent threshold defined by the original study [12]. For Freq, prenylation sequences scoring greater than -1 were deemed positive predictions, while sequences scoring greater than 0 were deemed positive predictions for cleavage [22].

## Experimental validation

**Yeast strains.** Strains used in this study are listed in S3 Table and are available upon request. Lithium acetate-based transformation methods were used to introduce plasmids into yeast strains [22, 34]. All strains were propagated at 25˚C unless otherwise stated, in YPD or appropriate selection media. For yWS2393, deletion of *STE24* was carried out in strain yWS44 (*mfa1Δ mfa2Δ*) using: a DNA fragment from pWS405 (*CEN URA3 ste24::KanMX4)* that was transformed into yWS44 [35]. G418 resistant colonies were checked by PCR for integration of *ste24*::*KANMX4* at the *STE24* locus. For yWS2462, deletion of *RCE1* was carried out in strain yWS44 using a *rce1*::*KAN* fragment recovered by PCR from the haploid yeast gene deletion collection [36], and integration at the *RCE1* locus was confirmed by PCR.

**Plasmids.** Plasmids used in this study are listed in S4 Table and are available upon request. All plasmids newly created for this study were constructed using methods previously reported [19, 22, 37]. Briefly, new plasmids encoding Ydj1p or **a**-factor reporters were constructed using PCR-directed recombination. Mutagenic oligonucleotides (S5 Table) encoding desired Cxxx sequences were co-transformed with linearized or gapped parent plasmids, transformation mixes plated onto appropriate selection media, and plasmids recovered from surviving colonies. Plasmids were sequenced through the entire open reading frame of the reporter using an appropriate DNA sequencing primer and a sequencing service (Genewiz, Southfield NJ; Eurofins Genomics, Louisville, Kentucky). pWS130 (*2μ URA3 P_{PGK}-HsRce1Δ22*) was constructed by subcloning a PCR-derived fragment from a baculovirus expression vector encoding *HsRce1Δ22* (courtesy of P. Casey, Duke University). The PCR fragment was designed to contain 5′ BamHI and 3′ PstI sites that were used for subcloning, where the latter was blunted with T4 Polymerase prior to cloning into the BamHI and SacII sites of pWS28 (2μ URA3 PPGK) [38]. pWS1609 was created from pWS1275 (*2μ URA3 P_{PGK}-HA-HsSTE24*) by PCR-directed, plasmid-based recombination to eliminate the HA-tag, followed by subcloning P_{PGK}-*HsSTE24* into pRS316 (*CEN URA3*) [37, 39].

**Ydj1p gel shift assay.** The prenylation status of Ydj1p was examined as described previously. Briefly, yeast strains expressing Ydj1p were cultured to A_{600} 0.9–1.1 at 30˚C in synthetic complete media lacking uracil (SC-U). Cell pellets of the same mass were collected by centrifugation, washed with water, and cell extracts prepared by alkaline hydrolysis followed by TCA precipitation [40]. Cell extracts were resuspended in Sample Buffer (250 mM Tris, 6 M Urea, 5% β-mercaptoethanol, 4% SDS, 0.01% bromophenol blue, pH 8) and analyzed by SDS-PAGE and immunoblotting with rabbit anti-Ydj1p antibody (courtesy of Dr. Avrom Caplan) and HRP conjugate antibody in TBST (10 mM Tris, 150 mM NaCl, 0.1% Tween-20; pH 7.5) with 1% milk/TBST. Blots were developed with WesternBright ECL Spray (Advansta Inc, San Jose, California), and images captured using X-ray film or a digital imager (Kwikquant, Kindle Biosciences, Greenwich, Connecticut). Prenylation was evaluated by quantifying Ydj1p band intensities using NIH ImageJ and calculating ratios for both prenylated and non-prenylated species or an equivalent gel position when a band was not apparent.

**Yeast mating assay.** Mating assays were performed as previously described [22]. Briefly, *MAT***a** and *MAT*α strains were cultured to saturation at 30˚C in synthetic complete media lacking leucine (SC-L) and YPD, respectively, then normalized to an A_{600} value of 1 by dilution with appropriate sterile media. *MAT***a** cultures were mixed individually 1:10 with the *MAT*α cultures, each mixture was serially diluted 10-fold using the normalized *MAT*α culture as the diluent, and serial dilutions were pinned onto minimal (SD) and synthetic complete media lacking lysine plates (SC-K). Plates were incubated for 72 hours and imaged against a black background using a flat-bed scanner. Images were adjusted using Photoshop to optimize the dynamic range of signal by adjusting input levels to a fixed range of 25–150.

# Results

## Prenylated and cleaved Cxxx sequences can be distinguished based on primary amino acid sequence feature

To evaluate whether the information encoded in primary sequences can be used to distinguish prenylated and cleaved sequences, we first curated a training dataset from two previously published genetic screens that used Ras61 and Ydj1p as reporters (Fig 1B) [8, 22]. As prenylation is necessary for the optimal function of both Ras61 and Ydj1p reporter activities, we curated 489 prenylated sequences by combining the top performing sequences from both screens. Another 508 low performing sequences from the Ras61 study served as the non-prenylated set;

the Ydj1p-based study did not yield information for low-performing sequences. Notably, prenylation and proteolysis have historically been considered coupled events, and as such, previous methodologies do not report on proteolysis. However, the Ydj1p reporter is uniquely able to differentiate between shunted (i.e., only prenylated) and cleaved sequences (i.e., canonically modified; prenylated, cleaved and carboxymethylated). Thus, we curated 136 sequences from the Ydj1p screen and 140 sequences from the Ras61 screen to serve as shunted and cleaved sets, respectively [22].

We next evaluated the contribution of three sequence representation methods: one hot encoding of primary sequence (sequence-only), AAindex, and ESM-1b. These methods capture different aspects of Cxxx sequences (see Materials & methods for additional details) in classifying prenylated and non-prenylated sequences. Two-dimensional projections of each set of features revealed that sequence-only and AAindex features readily distinguish prenylated and non-prenylated sequences, while ESM-1b exhibited poor separation (Fig 2A). As AAindex appeared to best separate the prenylated and non-prenylated sequences, we used Weblogo to analyze the sequences clustered with the right and left sides of the projection (Fig 2B). The right-side cluster was mostly composed of prenylated sequences that closely resembled the canonical definition of CaaX, with a clear enrichment of aliphatic amino acids at the $a_2$ position, and to some extent the $a_1$ position. By comparison, the left-side cluster was a mixed population of prenylated and non-prenylated sequences lacking these canonical aliphatic residues. Although ESM-1b encodes more information (276 dimensions to capture 99% variance in data compared to 50 dimensions for sequence and AAindex (see Materials & methods), the poor separation observed with ESM-1b is likely a consequence of the additional contextual information which could not be sufficiently compressed into two-dimensional space. All three sequence representation methods, meanwhile, are suitable for separating cleaved and uncleaved sequences (Fig 2C).

## SVM-ESM-1b outperforms several machine learning-based models for prenylation and cleavage predictions

A position-specific scoring matrix model (PSSM) is a common bioinformatics method employed for motif detection [41]. A variation of this method is used by the PrePS model [6]. We thus constructed a PSSM model based on the Cxxx sequences from our curated datasets to establish a baseline for comparisons of other prenylation and cleavage prediction models. The PSSM model applied to a curated dataset of both canonical and non-canonical sequences achieved 83.8 ± 3.3% accuracy for prenylation predictions, and a second PSSM model to predict cleavage achieved 93.8 ± 4.6% accuracy, based on 10-fold cross validation (Table 1). We next evaluated whether the baseline PSSM classification accuracy could be improved through different representations of Cxxx sequences using machine learning (see Materials & methods for details on methods used).

For prenylation, most of the 12 machine learning methods evaluated scored above 80% in all categories, with the variation in output from different machine learning algorithms being attributed to the different inductive biases used by each algorithm to make predictions about unseen data [42]. We selected the best model based on F1-score, defined as the harmonic mean of precision and recall. Based on this criterion, support vector machine (SVM) paired with ESM-1b features was the best overall performer. We next evaluated how well each model predicted prenylation of a validation set of 31 Cxxx sequences that were not part of training sets (S1 Table). Within this validation set, there was a subset of sequences found naturally within the yeast proteome, including 6 pairs (12 sequences) where the Cxxx motif differs by only one amino acid, and yet this change of one residue resulted in opposite prenylation

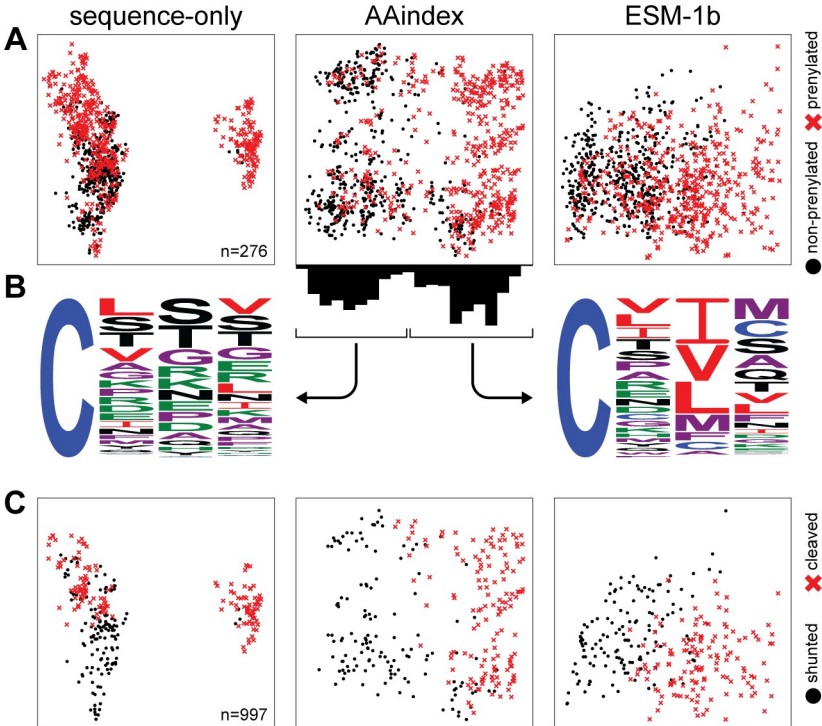

**Fig 2. Separation of sequences by machine learning-based methods.** A) Data points from all three features sets: sequence only, AAindex and ESM-1b, are represented as a two-dimensional projection of prenylated (red x) and non-prenylated sequences (black dot). The axes are not shown as they represent a linear combination of all features that maximizes variance. B) Bimodal distribution of sequences across the X-axis from the AAindex manifold were graphed as sequence logos. The distribution shown on the left contains a mix of non-prenylated Cxxx sequences and prenylated, non-canonical sequences, while the one on the right mostly consists of prenylated, canonical CaaX sequences. C) A similar two-dimensional projection was used to represent cleaved (red x) and shunted (i.e., uncleaved) sequences (black dot).

predictions across varying prediction methods. For example, CIIS (found on Ras2) is predicted to be prenylated while CIKS (Hmg1) is predicted to be non-prenylated. Seven additional, naturally occurring sequences were evaluated for various reasons–they were non-canonical and/or had varying prenylation or cleavage predictions across the different prediction methods (e.g., PrePS, Freq, etc.). In sum, 19 sequences from the yeast proteome were used for validation purposes. The remaining 12 sequences were chosen due to differing predictions by SVM-ESM-1b, PrePS, and the frequency-based scoring system (Freq). The 31 sequences representing the validation set were incorporated onto Ydj1p and prenylation evaluated by a gel shift assay (Fig 3A and 3B), with the exception of one sequence (CQSQ) that had been previously evaluated [22]. The effect of prenylation on protein migration by SDS-PAGE is not fully understood, but in the case of Ydj1p, it has long been established that prenylation causes a downward shift (i.e., faster migration) under carefully optimized gel conditions [20]. Cleavage does not impact Ydj1p mobility but does seem to impact the mobility of other proteins such as Ras [19, 22, 43]. Relative to PSSM, most machine learning methods improved at predicting actual prenylation (Table 1; Validation score). SVM was repeatedly the best overall performer when paired with ESM-1b features. Considering the results of performance testing with training and naïve test sets, SVM paired with ESM-1b features was chosen as the preferred machine learning method for additional prenylation prediction studies.

**Table 1. Performance of various models for prenylation prediction.**

| Model[a] | Features[b] | Accuracy[c] | Precision | Recall | F1 | Validation[d] |
|---|---|---|---|---|---|---|
| PSSM | sequence | 83.8 ± 3.3 | 87.7 ± 3.5 | 77.9 ± 5.9 | 82.4 ± 3.8 | 68.4 (13/19) |
| SVM | sequence | 86.0 ± 2.7 | 86.5 ± 4.0 | 84.9 ± 3.8 | 85.6 ± 2.8 | 84.2 (16/19) |
| SVM | AAindex | 85.1 ± 3.5 | 86.6 ± 4.1 | 82.4 ± 3.6 | 84.4 ± 3.6 | 73.7 (14/19) |
| SVM | ESM-1b | 86.4 ± 3.0 | 86.6 ± 3.3 | 85.5 ± 4.1 | 86.0 ± 3.1 | 84.2 (16/19) |
| GBDT | sequence | 86.2 ± 2.4 | 87.9 ± 3.5 | 83.4 ± 3.5 | 85.5 ± 2.6 | 68.4 (13/19) |
| GBDT | AAindex | 86.2 ± 2.8 | 87.2 ± 3.5 | 84.3 ± 4.3 | 85.6 ± 3.0 | 73.7 (14/19) |
| GBDT | ESM-1b | 85.0 ± 2.9 | 85.8 ± 3.8 | 83.2 ± 3.6 | 84.4 ± 3.0 | 78.9 (15/19) |
| Näive Bayes | sequence | 82.9 ± 1.8 | 85.5 ± 3.1 | 78.7 ± 3.3 | 81.9 ± 1.9 | 63.2 (12/19) |
| Näive Bayes | AAindex | 82.1 ± 3.0 | 82.2 ± 4.0 | 81.4 ± 3.6 | 81.7 ± 3.0 | 73.7 (14/19) |
| Näive Bayes | ESM-1b | 73.2 ± 2.3 | 70.4 ± 1.9 | 78.3 ± 3.9 | 74.1 ± 2.5 | 57.9 (11/19) |
| kNN | sequence | 84.1 ± 3.7 | 82.7 ± 4.3 | 85.5 ± 4.3 | 84.0 ± 3.7 | 78.9 (15/19) |
| kNN | AAindex | 82.7 ± 2.3 | 83.5 ± 3.1 | 81.0 ± 3.0 | 82.2 ± 2.3 | 78.9 (15/19) |
| kNN | ESM-1b | 83.0 ± 2.3 | 82.4 ± 3.5 | 83.4 ± 2.4 | 82.9 ± 2.1 | 78.9(15/19) |

[a]PSSM–Position-specific Scoring Matrix; SVM–support vector machine; GBDT–GradientBoost Decision Tree; kNN–k-Nearest Neighbors.

[b]Features for predicting sequence prenylation were based on one-hot encoding (sequence), physico-biochemical properties of amino acids (AAindex), and the ESM-1b Transformer model (ESM-1b).

[c]Reported percentages indicate the mean across 10-fold cross validation, while confidence intervals indicate the standard deviation.

[d]Reported percentages based off validation set tested *in vivo*

We also explored sequence cleavage using similar methods (Table 2; S2 Table). All models performed comparably well based on 10-fold cross validation, with most scoring above 90% in all categories. As observed for prenylation prediction, many of the models surpassed the PSSM model for accuracy and recall, and only 1 bettered PSSM for precision (Table 2). Overall, SVM

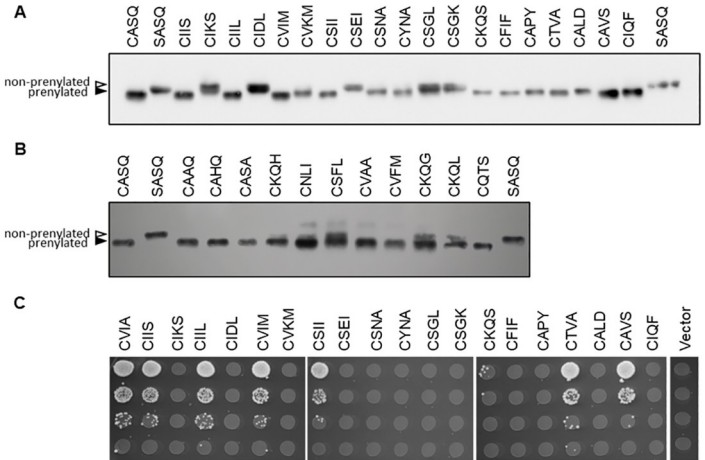

**Fig 3. Empirically determined prenylation and cleavage of various Cxxx sequences.** Yeast strains lacking chromosomally encoded *YDJ1* (yWS304 or yWS2544, *ydj1Δ*) or *MFA1* and *MFA2* (SM2331, *mfa1Δ mfa2Δ*) were engineered to individually express the indicated Ydj1p-Cxxx or **a**-factor-Cxxx variant, respectively, using a plasmid-based expression system. Prenylation of the indicated naturally occurring Cxxx sequences in yeast (A) or global Cxxx sequences (B) were determined by Ydj1p-gel shift assay. Yeast extracts were evaluated by SDS-PAGE and anti-Ydj1p immunoblot to reveal prenylated (closed triangle) and non-prenylated sequences (open triangle). Partial prenylation (i.e., doublet bands) were counted as a positive result. C) Cleavage of the indicated Cxxx sequences was determined by the **a**-factor mating assay. *MAT***a** yeast cultures were serial diluted 10-fold in the presence of excess *MATα* yeast (IH1793) and plated on SD media. Mating is indicated by diploid growth and is reported relative to mating exhibited by wildtype **a**-factor (CVIA).

**Table 2. Performance of various models for cleavage prediction.**

| Model[a] | Features | Accuracy | Precision | Recall | F1 | Validation |
|---|---|---|---|---|---|---|
| PSSM | sequence | 93.8 ± 4.6 | 97.1 ± 4.5 | 90.7 ± 7.9 | 93.6 ± 4.9 | 89.4 (12 / 14) |
| SVM | sequence | 97.5 ± 2.3 | 96.7 ± 4.3 | 98.6 ± 2.9 | 97.5 ± 2.2 | 78.9 (10 / 14) |
| SVM | AAindex | 96.4 ± 2.8 | 95.3 ± 4.1 | 97.9 ± 3.3 | 96.5 ± 2.7 | 78.9 (10 / 14) |
| SVM | ESM-1b | 97.5 ± 1.6 | 97.3 ± 3.3 | 97.9 ± 3.3 | 97.5 ± 1.6 | 78.9 (10 / 14) |
| GBDT | sequence | 94.9 ± 3.4 | 94.6 ± 3.9 | 95.8 ± 5.7 | 95.0 ± 3.4 | 52.6 (8 / 14) |
| GBDT | AAindex | 86.9 ± 3.2 | 87.7 ± 4.3 | 85.3 ± 3.1 | 86.4 ± 3.2 | 73.7 (11 / 14) |
| GBDT | ESM-1b | 86.2 ± 1.9 | 87.0 ± 2.8 | 84.5 ± 2.3 | 85.7 ± 1.9 | 78.9 (10 / 14) |
| Näive Bayes | sequence | 89.9 ± 6.0 | 89.1 ± 6.6 | 91.5 ± 7.0 | 90.1 ± 5.9 | 68.4 (9 / 14) |
| Näive Bayes | AAindex | 94.2 ± 2.4 | 94.5 ± 4.0 | 94.3 ± 4.3 | 94.3 ± 2.3 | 89.4 (12 / 14) |
| Näive Bayes | ESM-1b | 85.5 ± 7.6 | 85.3 ± 7.3 | 86.4 ± 10.3 | 85.7 ± 7.9 | 68.4 (9 / 14) |
| kNN | sequence | 94.9 ± 4.3 | 96.5 ± 4.7 | 93.6 ± 5.9 | 94.9 ± 4.4 | 84.2 (12 / 14) |
| kNN | AAindex | 94.6 ± 3.3 | 92.4 ± 5.9 | 97.9 ± 3.3 | 94.9 ± 3.0 | 78.9 (10 / 14) |
| kNN | ESM-1b | 95.3 ± 4.5 | 94.7 ± 5.6 | 96.4 ± 5.8 | 95.4 ± 4.5 | 78.9 (10 / 14) |

[a]Terms, definitions, and calculations are as described for Table 1.

paired with either sequence or ESM-1b features achieved the best F1-score for predicting cleavage. As SVM-ESM-1b had the smaller standard deviation, it was chosen as the preferred method for cleavage prediction. We next evaluated how well each model predicted cleavage of the validation set of 19 naturally occurring Cxxx sequences. We incorporated these 19 sequences onto the **a**-factor reporter that conditionally requires both prenylation and cleavage for bioactivity (Fig 3C). Because 5 of the sequences were not observed to be prenylated by gel-shift assay (CIKS, CIDL, CSEI, CSGL, CSGK), these sequences were not expected to exhibit any **a**-factor activity, which was indeed the case. For this reason, these 5 sequences were not included statistically in the **a**-factor validation set (i.e., CaaX cleavage is prenyl-dependent). The remaining 14 sequences either possessed **a**-factor activity, indicative of cleavage, or lacked bioactivity, indicative of only being prenylated. Surprisingly, we found that several models out-performed SVM-ESM-1b on the validation set when considering the 14 prenylated sequences (Table 3, S1 Table). We caution, however, that the small size of the validation set may lack sufficient statistical power to make proper comparisons and conclusions.

## Global predictions for prenylation and cleavage of Cxxx sequence space

After evaluating different models for prenylation and cleavage with our curated training and validation sets, we chose SVM paired with ESM-1b to predict both prenylation and cleavage for the full scope of Cxxx sequences (S2 File). In the case of prenylation, our model was trained to make binary predictions, but these sequence predictions are better represented on a continuum as partial prenylation could occur, resulting in sequences with fractions of the protein population being prenylated. In order to model this continuum, we obtained probabilistic outputs for the SVM model by Platt scaling [28] (Fig 4). We note that this method only provides an estimated probability, which does not perfectly translate to a strict cutoff value for the actual binary classification. Altogether, our analysis of all 8000 Cxxx sequences predicts that 67% (n = 5373) are unmodified, 18% (n = 1420) are shunted (i.e., prenylation only), and 15% (n = 1217) cleaved (i.e., canonically modified; prenylated, cleaved, and carboxylmethylated) (Fig 5A and 5D). We also made global predictions using the SVM-ESM-1b prenylation model paired with our previously published Freq method that outperformed all machine learning models on cleavage validation score (Fig 5B and 5E), as well as using Freq for both prenylation

**Table 3. Comparison of prenylation and cleavage prediction models with empirical observations.**

| | yeast protein | CaaX | Prenylation | | | | | Cleavage | | |
| | | | SVM[a,b] | PrePS | Freq | FPB | Observed[c] | SVM[a] | Freq | Observed[d] |
|---|---|---|---|---|---|---|---|---|---|---|
| **similar sequences** | Ras2 | CIIS | + | + | + | + | + | + | + | + |
| | Hmg1 | CIKS | - | - | - | - | - | NA | NA | NA |
| | Rho2 | CIIL | + | + | + | - | + | + | + | + |
| | Ssp2 | CIDL | - | - | - | - | - | NA | NA | NA |
| | Skt5, MiY1 | CVIM | + | + | + | - | + | + | + | + |
| | Tbs1 | CVKM | - | - | - | - | + | + | - | - |
| | YDL022C-A | CSII | + | + | + | + | + | + | + | + |
| | YBR096W | CSEI | - | - | - | - | - | NA | NA | NA |
| | YMR265C | CSNA | - | - | + | - | + | - | - | - |
| | Pet18 | CYNA | - | - | - | + | + | - | - | - |
| | Lih1 | CSGL | - | - | + | - | - | NA | NA | NA |
| | Cup1 | CSGK | - | - | - | - | - | NA | NA | NA |
| **other sequences** | Nap1 | CKQS | + | + | + | - | + | - | - | - |
| | Cst26 | CFIF | + | + | - | - | + | + | - | - |
| | YIL134C-A | CAPY | + | + | - | - | + | - | - | - |
| | Atr1 | CTVA | + | + | + | + | + | + | + | + |
| | Las21 | CALD | + | - | + | + | + | - | + | - |
| | YDL009C | CAVS | + | + | + | + | + | - | + | + |
| | Sua5 | CIQF | + | + | + | - | + | + | - | - |
| | **number observed/predicted** | | 16/19 | 15/19 | 14/19 | 11/19 | | 10/14 | 13/14 | |

[a]Signs represent predictions of prenylation and cleavage that were reported as positive (+) or negative (-) by the indicated model. NA—not applicable; the non-prenylated status of the sequence precludes it from being cleaved; CaaX cleavage is prenyl-dependent.

[b]SVM–SVM-ESM-1b; PrePS–Prenylation Prediction Suite; Freq–in-house, frequency-based; FPB–FlexPepBind.

[c]Observed by Ydj1p prenylation gel shift–see Fig 3A, S1 Fig.

[d]Observed by **a**-factor mating–see Fig 3C.

predictions and cleavage (Fig 5C and 5F) [22]. All predictions were qualitatively similar, with the majority of the 8000 sequences being unmodified, and more shunted sequences predicted relative to canonical sequences.

## Comparisons to previous prenylation methods and evaluation of yeast proteome predictions

Several prenylation predictors have been developed previously. These include: PrePS, a PSSM-based model; FlexPepBind (FPB), a molecular docking-based model encompassing energy scores; and Freq, an in-house method developed by scoring the frequency of residues at each position in the positive and negative testing sets used for machine learning in this study [6, 12, 22]. Relative to all 8000 Cxxx sequence space, our SVM-ESM-1b based model predicts prenylation for more sequences (33%) in comparison to PrePS (20%) and FlexPepBind (17%), but less by comparison to Freq (42%) (S2 File). While Freq predicts more prenylated sequences, it is important to note that this method overpredicts prenylation in the negative training set relative to the SVM-ESM-1b model (~40% vs. 3%, respectively). A potential explanation for the higher false positive rate of Freq may be that this method does not explicitly encode contextual information when generating features. Overall, we conclude that the SVM-ESM-1b based machine learning model predicts more prenylatable space, as compared to PrePS and

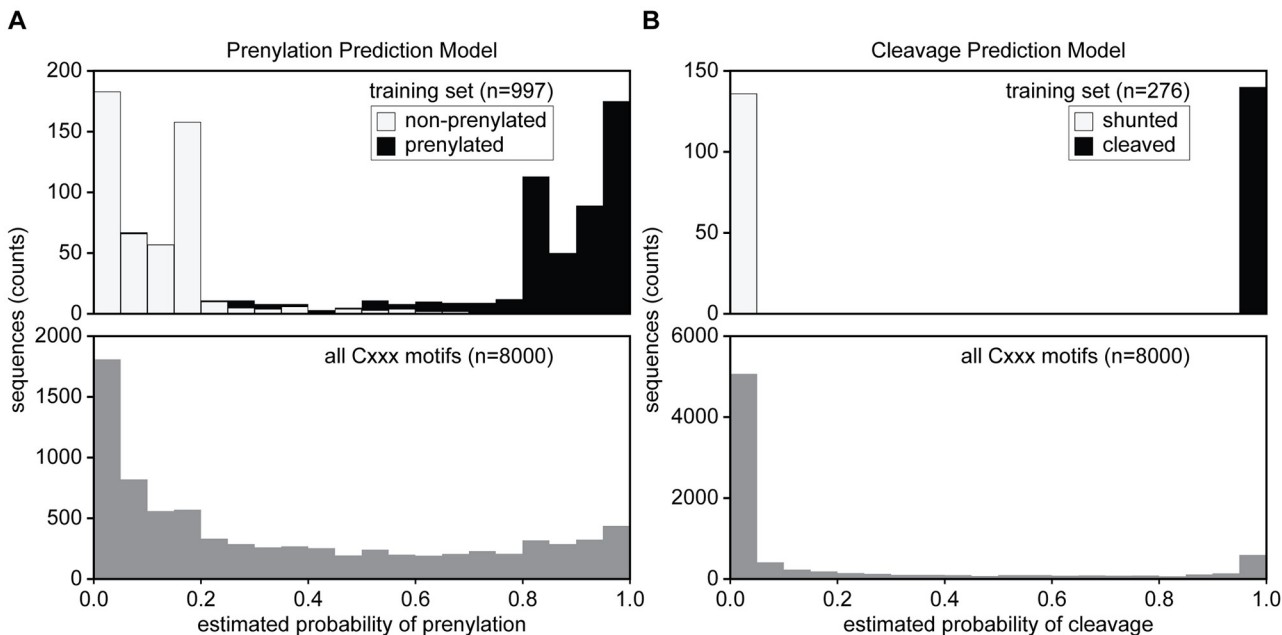

**Fig 4. Probability distributions for prenylation and cleavage predictions made by SVM-ESM-1b.** Probability distributions for both prenylation (A) and cleavage (B) determined for the training sets (top) and for all 8000 Cxxx motifs (bottom). A) For prenylation, the training set distribution is represented as a stacked bar plot where prenylated sequences are black, while non-prenylated sequences are white. B) For cleavage, the training set distribution is represented as a stacked bar plot where shunted sequences (prenylation only) are white and cleaved sequences for proteolysis are black. The probability distributions were determined for the training sets (top) and for all 8000 Cxxx motifs (bottom).

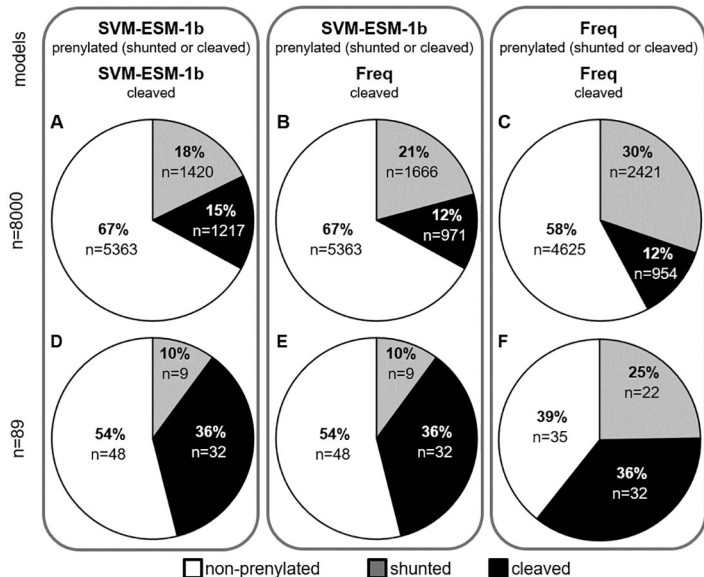

**Fig 5. Predictions for modification of Cxxx sequences based on various methods.** Predictions for prenylation and cleavage for all 8000 Cxxx sequences (A-C) and 89 naturally occurring yeast Cxxx sequences (D-F). Models used for prenylation prediction were SVM-ESM-1b (A,B,D,E) and the in-house, frequency-based method (C,F). Models used for cleavage prediction were SVM-ESM-1b (A,D) and the in-house, frequency-based method (B,C,E,F). Predictions are binned as non-prenylated (white), shunted (gray), and cleaved sequences (black). Cleaved sequences have a prerequisite of being prenylated by the indicated prediction model.

FlexPepBind, and may more accurately predict prenylation than our previously reported Freq method. Regarding CaaX cleavage prediction, Freq has been the only available method for binning prenylated sequences as either shunted or cleaved. Freq predicts more shunted sequences relative to PSSM-based predictions (30% vs. 21%, respectively), while the prediction for cleaved sequences is the same in both cases (12%) (Fig 5B and 5C).

Altogether, the yeast genome contains 89 proteins having Cxxx at the COOH-terminus. Prenylation and cleavage predictions were determined for the Cxxx sequences associated with these proteins using our SVM-ESM-1b and PSSM models, respectively. SVM predicted 41 yeast Cxxx proteins to be prenylated, where 32 were canonically modified and 9 were shunted (Fig 5B). While many of the canonically modified CaaX proteins have been previously characterized (**a**-factor, Ras, etc.), some have non-canonical Cxxx sequences and have not been previously evaluated for their prenylation status, including Cst26p (CFIF; an acyltransferase) and Sua5p (CIQF; involved in threonylcarbamoyladenosine synthesis). Of the 89 Cxxx sequences associated with the yeast proteome, 19 were directly evaluated in this study in the context of the Ydj1p reporter (Table 3, Fig 3A, S1 Fig). The SVM-ESM-1b model correctly predicted the prenylation (both positive and negative) for 84% of the sequences. By comparison, PrePS was next best, correctly predicting 79%, followed by Freq correctly predicting 74%, and FPB correctly predicting 58%. Because SVM-ESM-1b, PrePS, and Freq performed similarly in predicting prenylation of naturally occurring Cxxx sequences, we evaluated additional sequences to better differentiate the prediction methods. Our lab possesses a large collection of plasmids encoding Ydj1-Cxxx variants (n > 200). Excluding those with Cxxx sequences that were part of machine learning training sets and others for which SVM-ESM-1b and PrePS had the same prediction led us to 12 plasmids with varying differential predictions by SVM, PrePS, and Freq (Table 4). For these 12 Cxxx sequences, Freq correctly predicted 10, SVM-ESM-1b correctly predicted 9, and PrePS correctly predicted 4 (Table 4, Fig 3B). Evaluation of immunoblot band intensities revealed that all 12 sequences were either fully prenylated (100% prenylation, n = 7) or mostly prenylated (>75% prenylation, n = 5), indicating a higher percentage of false negatives for PrePS relative to other prediction models. Thus, for the combined set of 31 sequences evaluated for prenylation, SVM correctly predicted 81% (25/31), Freq correctly identified 77% (24/31), and PrePS correctly predicted 61% (19/31) (Table 5).

For assessing cleavage, we used the yeast **a**-factor mating pheromone as a reporter (Fig 3C). Canonical modification of **a**-factor (i.e., prenylation, cleavage, and carboxylmethylation) is required for mating of haploid yeast, which can be quantified as an indirect measure of **a**-factor production. As noted previously, CaaX cleavage is prenyl-dependent, so we only evaluated the 14 sequences that were confirmed as being prenylated by Ydj1p gel-shift, regardless of whether they were predicted to be prenylated by any computational method. In this case, Freq outperformed SVM-ESM-1b, correctly predicting cleavage for 93% of sequences compared to 71%, respectively; FBP and PrePS are not able to predict cleavage, so they were not evaluated (Table 3). For sequences where mating is observed, the mating levels are comparable to that of the wild type **a**-factor sequence (CVIA) (Fig 3C), indicative of complete rather than partial cleavage.

## Limitations of machine learning for predicting CaaX protein PTMs

While SVM-ESM-1b can predict prenylation and cleavage, one limitation is that it does not provide any information about enzyme specificity due to the lack of enzyme-specific training information. For both prenylation and proteolysis, there are two possible enzymes for each reaction. For prenylation, FTase and GGTase-I can each prenylate a wide array of CaaX proteins with C15 farnesyl and C20 geranylgeranyl, respectively, while for proteolysis, Rce1p and

**Table 4. Comparison of SVM-ESM-1b and PrePS prenylation predictions with empirical observations.**

| Reporter | Prenylation | | | | |
|---|---|---|---|---|---|
| Ydj1-Cxxx | SVM[a,b] | PrePS | Freq | FPB | Observed[c] |
| CAAQ | + | - | + | - | + |
| CAHQ | + | - | + | - | + |
| CASA | + | - | + | - | + |
| CKQH | + | - | + | - | + |
| CNLI | + | - | + | - | + |
| CSFL | + | - | + | - | + |
| CVAA | + | - | + | - | + |
| CVFM | + | - | + | - | + |
| CKQG | - | + | + | - | + |
| CKQL | - | + | + | - | + |
| CQTS | - | + | - | - | + |
| CQSQ[d] | + | + | - | - | + |
| **number observed/predicted** | 9/12 | 4/12 | 10/12 | 0/12 | |

[a]Signs represent predictions of prenylation and cleavage that were reported as positive (+) or negative (-) by the indicated model.
[b]SVM–SVM-ESM-1b; PrePS–Prenylation Prediction Suite; Freq–in-house, frequency-based; FPB–FlexPepBind.
[c]Observed by Ydj1p prenylation gel shift–see Fig 3B.
[d]Observation previously reported [22].

Ste24p are both able to cleave the farnesylated CVIA motif of **a**-factor, but selectivity is observed for other motifs. The determinants of substrate specificity have not been fully ascertained for the aforementioned enzymes. A case in point is proteolysis of the CaaX motif CSIM, a sequence found on human prelamin A that has long thought to be a substrate of both CaaX proteases. SVM-ESM-1b and PSSM both predict that CSIM is cleaved, which we confirmed by using the **a**-factor reporter. When both proteases were present, comparable mating levels were observed between strains expressing **a**-factor with the native CVIA motif that is cleaved by both Rce1p and Ste24p, the CTLM motif that is Rce1p-specific, and the CSIM motif for which cleavage specificity is debated (Fig 6A). When evaluated in the context of just one CaaX protease, we observed that all three motifs could be cleaved by Rce1p, but only CVIA was cleaved by Ste24p (Fig 6B). A similar result was observed when evaluating cleavage of these sequences by the human CaaX proteases expressed in our yeast system (Fig 6C). Our observations are consistent with multiple reports challenging the role of Ste24p as an authentic CaaX protease,

**Table 5. Summary of prenylation and cleavage predictions.**

| | Prenylation | | | | Cleavage | |
|---|---|---|---|---|---|---|
| | SVM[a,] | PrePS | Freq | FPB | SVM[a] | Freq |
| **number observed/predicted**[b] | **25/31** | **19/31** | **24/31** | **11/31** | **10/14** | **13/14** |
| **%observed/predicted** | 81% | 61% | 77% | 28% | 71.4% | 92.9% |
| **number false positive** | 0/20 | 0/14 | 1/22 | 0/6 | 3/7 | 1/7 |
| **% false positive** | 0 | 0 | 4.5% | 0 | 42.8% | 14.3% |
| **number false negative** | 6/11 | 12/17 | 5/9 | 20/25 | 1/7 | 0/7 |
| **% false negative** | 54.5% | 70.6% | 55.5% | 80% | 14.3% | 0 |

[a] SVM–SVM-ESM-1b; PrePS–Prenylation Prediction Suite; Freq–in-house, frequency-based; FPB–FlexPepBind.
[b]Values determined by empirical data via Ydj1p prenylation gel shift (prenylation, Fig 3A and 3B, S1 Fig) or **a**-factor mating (cleavage, Fig 3C).

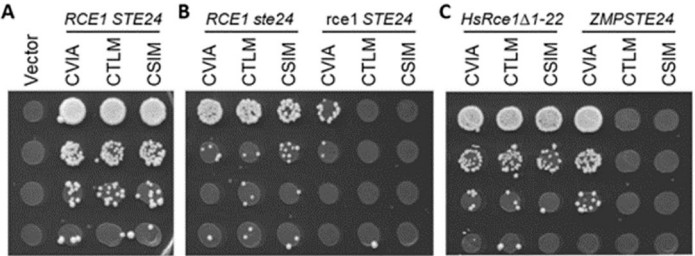

**Fig 6. Rce1 is responsible for cleavage of yeast a-factor-CSIM.** Yeast strains expressing the indicated **a**-factor Cxxx variant as the sole source of **a**-factor were evaluated as described for Fig 3 in the context of yeast and human CaaX proteases. Yeast strains expressing A) both yeast CaaX proteases (SM2331, *mfa1Δ mfa2Δ*), B) one or the other yeast CaaX protease (yWS2393, *mfa1Δ mfa2Δ ste24*; yWS2462, *mfa1Δ mfa2Δ rce1*), or C) plasmid-based human CaaX proteases (pWS130, *HsRce1Δ1–22*; pWS1609, *ZMPSTE24*) in a strain lacking both yeast CaaX proteases (yWS164, *mfa1Δ mfa2Δ rce1 ste24*).

including a recent *in vitro* study demonstrating the inability of the human Ste24 ortholog, ZMPSTE24 to cleave at the Cys(farnesyl)-Ser bond of the CSIM motif, as would be expected for a CaaX protease [44].

## Discussion

A collection of *in vivo*, *in silico* and *in vitro* observations support a wider array of prenylation substrates than those previously defined by the COOH-terminal CaaX motif [8–12, 22]. Among the new substrates are those that lack aliphatic amino acids at the $a_1$ and $a_2$ position, leading to a broader definition for the prenylation motif. To develop a robust machine learning platform to predict prenyation, we first tested three sequence representation methods. Initially, AAindex appeared to best distinguish prenylated versus non-prenylated sequences, with ESM-1b appearing to show poor separation (Fig 2A). The reason for these differences in the feature representation methods is likely because AAindex directly utilizes known biophysical features while ESM-1b utilizes a learned embedding vector based on biologically observed sequences. Because ESM-1b features are a product of representation learning, they cannot be directly translated into concrete biophysical features. For prenylation, both AAindex and ESM-1b features were good at modeling the preference for aliphatic residues in the Cxxx motif. Differences between the two methods emerge, however, when modeling non-canonical prenylation motifs. ESM-1b features uniquely predict Cxxx sequences containing Gln at x2 or x3 to be prenylated, while AAindex uniquely models a preference for Trp and Phe at x2. We speculate that AAindex features may incorrectly model a general preference for nonpolar residues, due to the enrichment of aliphatic residues in the training set.

After further training and cross-validation, it was determined that the machine learning platform SVM paired with ESM-1b training on CaaX motifs identified using both shunted and canonical reporters was the best overall performer in predicting canonical and non-canonical Cxxx prenylation. SVM-based predictions suggest that approximately 33% of all 8000 Cxxx motifs are prenylatable. This estimate is approximately 50% higher than the number of potential targets predicted by PrePS and is approximately double the number of sequences predicted by FlexPepBind (FPB). These findings are not meant to be indicative of the number of prenylated proteins in a cell since far fewer than all 8000 possible Cxxx motifs are encoded in genomes. For example, *S. cerevisiae* encodes only 89 proteins that end Cxxx. Of these, SVM-ESM-1b predicted 46% (n = 41) to be prenylated. By comparison, FPB and PrePS predicted 27% (n = 24) and 32% (n = 29) of yeast proteins to be prenylated, respectively. Confirmation of SVM-predicted prenylation will need to be evaluated on a case-by-case basis or by

application of emerging methods for *in vivo* labeling of prenylproteins to firmly establish whether SVM is an improvement over previous methods. Despite the potential limitations of our prediction method, it is clear that SVM-ESM-1b predicted prenylation of known, non-canonical Cxxx sequences in instances where other methods did not (e.g., Ydj1p CASQ and Pex19p CKQQ) (S2 File), suggesting that SVM is an improvement for identifying prenylated proteins as a whole. Moreover, the non-canonical CKQS sequence associated with the histone chaperone Nap1p is also predicted to be prenylated by our SVM-ESM-1b model. To date, there exists no direct evidence for yeast Nap1p prenylation, but such evidence does exist for human and plant Nap1 homologs, which both possess a similar CKQQ motif [45, 46]. Notably, the CKQQ sequence is also present on the human tumor suppressor Lkb1, another well documented prenylprotein [47].

As part of this study, we were also able to develop SVM-ESM-1b into a first-ever method for distinguishing between shunted (i.e., prenylation only) and cleaved sequences (i.e., canonical). Of the approximately 2600 sequences predicted to be prenylated by SVM, approximately 63% are predicted to be shunted and the remaining 37% cleaved. Again, these findings are not meant to reflect the actual ratio of shunted and cleaved prenylated proteins in cells. In fact, we observe that the predictions are somewhat inversed within the yeast proteome. Of the 41 sequences predicted to be prenylated, 27% are predicted to be shunted and the remaining 73% cleaved. This observation suggests that the cleavage and carboxymethylation of the prenylated COOH terminus may serve an important role *in vivo*, potentially increasing membrane association, as historically expected for canonical CaaX modifications. While the role of the isoprenyl group on shunted proteins remains unclear, we posit that this PTM may help mediate protein-protein interactions and/or provide a structural role rather than contribute to membrane association. This is supported by observations made on the human protein Spindly, whose CPQQ sequence is predicted to be shunted by our SVM model and for which a farnesyl-dependent protein complex interaction has been proposed [48, 49].

An unexpected result from this study was the observation that Freq and SVM-ESM-1b had a similar level of accuracy for prenylation prediction of the validation set (77% and 80%, respectively). As noted previously, Freq globally predicted more prenylated sequences than SVM-ESM-1b (42% and 33%, respectively), which is consistent with Freq having a higher false positive rate compared to SVM for our negative training set (40% and 3%, respectively). This suggests to us that Freq overpredicts prenylation. It's also worth noting that while Freq and SVM-ESM1b rely on the same data set for predictions, their predictions are not coincident, indicating that predictions are fundamentally different for the two methods. Long term, we expect that future advancements in machine learning will lead to better prediction performance relative to the Freq-based method.

To further improve our prediction methods, one aspect that we wish to especially improve upon is the high false negative rate for prenylation predictions that was determined empirically by evaluating a small subset of test sequences (n = 31; Table 5). While a larger test set may yield a more accurate false negative rate, it remains possible that the high negative false rate is simply due to the training datasets themselves being too small or somehow compromised. We have high confidence that our positive prenylation training set is composed of prenylated sequences that, importantly, were derived from studies involving both canonical and shunted reporters. Our negative training test set, however, was derived from a single study that relied on a canonical reporter, and it is suspected that shunted sequences may be among the negative hits in that study, thus poisoning the quality of our negative test set. Our future studies are aimed at identifying a set of sequences that better reflect non-prenylatable sequences for use as an improved negative training set that we expect to lead to improved prenylation predictions and a lower false negative rate. Ultimately, we expect that prenylation potential will be best

defined by knowing the profiles of both preferred and non-preferred residues. The reasons for both states will also need to be explored. One likely possibility may be that only certain individual residues or groups of residues can be accommodated in the active site of prenyltransferases. But, we also recognize that preferred residues may be insufficient to drive prenylation if the Cxxx sequence is buried in the protein core, is involved in secondary structure that is required for protein stability, is exposed on the protein surface but involved in stable interactions with partner proteins, is oriented topologically away from the cytosol and inaccessible to the cytosolic prenyltransferases (i.e., outside the cell or within organelles).

Interestingly, we observed that several models out-performed SVM-ESM-1b for cleavage prediction (e.g., PSSM, Freq). As previously noted, a larger set of test sequences may be needed to better assess performance. Alternatively, it may be that a better genetic test for cleavage is required. Previous studies have reported that geranylgeranylated **a**-factor has less mating activity *in vivo* [18, 22, 50], suggesting that the genetic mating assay may only work well in the context of farnesylated **a**-factor. This potentially impacts results associated with the CFIF and CIQF sequences in our test set; the terminal Phe is a preferred GGTase-I feature. SVM-ESM-1b predicted prenylation of both sequences while SVM-ESM-1b, PSSM and Freq methods all predicted cleavage. Prenylation was confirmed in the context of Ydj1p, but neither sequence supported **a**-factor mating activity that would be indicative of cleavage. It remains unclear whether lack of mating activity is due to shunting or geranylgeranylation. Because of this issue, it is difficult to fully assess the accuracy of any of the cleavage predictors described in this study. In terms of the CaaX proteases, while CSIM was identified as a canonical motif, additional genetic studies utilizing **a**-factor were needed to resolve whether cleavage was mediated by Rce1p or Ste24p. As the yeast **a**-factor mating pheromone is the only known substrate of Ste24p to date, it is tempting to speculate that Rce1p is the main and possibly only relevant CaaX protease. If that eventually bears out to be the case, then our cleavage predictors could be used to infer Rce1p specificity.

Altogether, we have demonstrated that machine learning can be developed into a useful tool to predict prenylation and cleavage events associated with CaaX proteins. The utility of this tool is reflected by its ability to better identify possible shunted sequences relative to other publicly available prediction methods, in addition to identifying canonically modified sequences. These findings represent an important step in expanding the full scope of prenylatable motifs in yeast. Given the high degree of target specificity exhibited by both prenyltransferases and CaaX proteases across species, it is likely that the prenylatable space identified by this study also represents the full scope of prenylated motifs in humans. Among these are sequences associated with proteins that represent potential new additions to the prenylome, which has implications for the impact of prenyltransferase and protease inhibitors being developed as therapeutics.

## Supporting information

**S1 Fig. Confirmation of prenylation status on ambiguous Cxxx sequences.** Yeast strains lacking chromosomally encoded *YDJ1* +/- *RAM1* (yWS304, *ydj1Δ* or yWS2542, *ydj1Δram1Δ*) expressing Ydj1p-Cxxx plasmids of sequences were evaluated in the presence/absence of FTase (*RAM1* gene) as described in Fig 3. Sequences were selected from Fig 3A for further evaluation due to unclear gel shift or prenylation status.
(TIF)

**S1 Table. Probability estimates and prediction calls for prenylation and cleavage of naturally occurring yeast Cxxx sequences as reported by the SVM-ESM-1b model.**
(DOCX)

**S2 Table. Prediction calls for cleavage of naturally occurring yeast Cxxx sequences by indicated model.**
(DOCX)

**S3 Table. Yeast strains used in this study.**
(DOCX)

**S4 Table. Yeast expression plasmids used in this study.**
(DOCX)

**S5 Table. PCR Oligonucleotides used in this study.**
(DOCX)

**S1 File. Training sets used for machine learning.**
(XLSX)

**S2 File. Prenylation and cleavage predictions for all Cxxx sequences.**
(XLSX)

**S1 Data.**
(ZIP)

**S1 Raw images. Raw blot images.**
(PDF)

## Acknowledgments

We thank Avrom Caplan (City College of New York) for anti-Ydj1p antibody, Patrick Casey (Duke University) for the baculovirus vector encoding human Rce1, and Ora Furman-Schueler (Hebrew University of Jerusalem) for sharing FlexPepBind scores. We also thank Jacob Greenway (Schmidt Lab, UGA) and members of the Schmidt Lab for their assistance with methods, reagent preparation, and critical discussions.

## Author Contributions

**Conceptualization:** Brittany M. Berger, Wayland Yeung, Natarajan Kannan, Walter K. Schmidt.

**Investigation:** Brittany M. Berger, Wayland Yeung, Arnav Goyal, Zhongliang Zhou.

**Methodology:** Brittany M. Berger, Wayland Yeung.

**Resources:** Emily R. Hildebrandt.

**Supervision:** Natarajan Kannan, Walter K. Schmidt.

**Writing – original draft:** Brittany M. Berger.

**Writing – review & editing:** Wayland Yeung, Arnav Goyal, Zhongliang Zhou, Emily R. Hildebrandt, Natarajan Kannan, Walter K. Schmidt.

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
