## [Decision Letter · Decision Letter 0]

6 Apr 2022

PONE-D-21-33646Functional classification and validation of yeast prenylation motifs using machine learning and genetic reportersPLOS ONE

Dear Dr. Schmidt,

Thank you for submitting your manuscript to PLOS ONE. After careful consideration, we feel that it has merit but does not fully meet PLOS ONE’s publication criteria as it currently stands. Therefore, we invite you to submit a revised version of the manuscript that addresses the points raised during the review process. The reviewers have made some minor suggestions to improve the manuscript which I believe should be straight forward to address. 

We look forward to receiving your revised manuscript.

Kind regards,

Patrick Lajoie, PhD

Academic Editor

PLOS ONE

https://journals.plos.org/plosone/s/file?id=ba62/PLOSOne_formatting_sample_title_authors_affiliations.pdf".

“This work was supported by NIH funds to WKS and NK (NIH NIGMS GM132606, https://www.nih.gov/) and funds to WKS (NIH NIGMS R01GM117148, https://www.nih.gov/).  The funders had no role in study design, data collection and analysis, decision to publish, or preparation of the manuscript.”

“This work was supported by NIH funds to WKS and NK (NIH NIGMS GM132606, https://www.nih.gov/) and funds to WKS (NIH NIGMS R01GM117148, https://www.nih.gov/).  The funders had no role in study design, data collection and analysis, decision to publish, or preparation of the manuscript.”

Reviewers' comments:

Reviewer's Responses to Questions

**Comments to the Author**

1. Is the manuscript technically sound, and do the data support the conclusions?

Reviewer #1: Yes

Reviewer #2: Yes

2. Has the statistical analysis been performed appropriately and rigorously? 

Reviewer #1: Yes

Reviewer #2: Yes

3. Have the authors made all data underlying the findings in their manuscript fully available?

Reviewer #1: Yes

Reviewer #2: Yes

4. Is the manuscript presented in an intelligible fashion and written in standard English?

Reviewer #1: Yes

Reviewer #2: Yes

5. Review Comments to the Author

Reviewer #1: Summary:

C-terminal CaaX motifs, such as those found in the Ras GTPase and the yeast a-factor mating pheromone, are canonically thought to direct cellular machinery to prenylate the cysteine side chain, cleave the aaX residues, and methylate the newly exposed cysteine carboxy terminus. This paper by Brittany M Berger and colleagues in Walter Schmidt’s lab follows up on the 2016 eLife study from the Schmidt lab, which revealed that not all CaaX proteins are modified in this canonical manner: the yeast J-domain protein/Hsp40 chaperone Ydj1 is prenylated but avoids classical downstream processing steps (carboxymethylation and cleavage), and this “shunted” modification is critical for Ydj1 to allow cells to grow at elevated temperature. Here, the authors test various machine learning algorithms in combination with various feature tabulation methods to attempt to classify which CaaX motifs direct proteins to the canonical pathway vs. the “shunted” pathway utilized by Ydj1. By using the non-canonical prenylation reporter Ydj1, they are able to expand the predicted set of prenylation sites beyond those whose final fate is carboxy methylation. The authors found multiple sequences (often, without the canonical aliphatic amino acids) that were prenylated but not cleaved. These sites would not be found by previous prenylation detection assays, which rely on cleavage as a readout of prenylation. With reporters of cleaved and non-cleaved prenylation targets in hand, the authors were also able to develop a cleavage prediction algorithm that performs well with their small training set. Unfortunately, the machine learning algorithm in this paper is limited in its predictive power because the training sets were small. As the authors acknowledge, this prevents a conclusive analysis and comparison of their machine learning algorithms and feature tabulation methods. However, the algorithm has improved predictive power, and coupled with the establishment of reporter assays for both the canonical and shunted pathways, this paper advances the field.

Specific comments:

1) The figures begin abruptly and it’s difficult to understand the first result. It would be helpful to include a cartoon figure at the beginning showing the canonical and shunted pathways and schematizing the workflow of the machine learning process.

2) The rationale for use of the 19-sequence validation set was unclear. The authors should justify more clearly why only these 19 sequences were used.

3) For the uninitiated reader, it would be helpful to explain why the prenylated proteins migrate faster in the gel than the unmodified proteins, and what to expect in terms of mobility for cleaved vs. shunted proteins.

4) Describe which features of Cxxx sequences are leading to the differences in the ESM-1b vs. Aaindex feature representation method results.

5) Figures are in reverse order in the manuscript and some figures are mislabeled.

6) Figure 1a: Label how many data points are graphed. n=489+508?

7) Fig 1c: Label how many data points are graphed. n=136+140

8) Page 5, Line 115 should also cite prenylation effect on Ydj1-Hsp90 interaction: Flom GA, Lemieszek M, Fortunato EA, Johnson JL. Farnesylation of Ydj1 is required for in vivo interaction with Hsp90 client proteins. Mol Biol Cell. 2008 Dec;19(12):5249-58. doi: 10.1091/mbc.e08-04-0435. Epub 2008 Oct 1. PMID: 18829866; PMCID: PMC2592663.

9) Figure legend 2B: cleaved vs shunted figure legend and bar colors are reversed. Shunted should have probability = 0 of being cleaved.

10) Figure legend and bar colors in Fig 2A are also reversed.

11) Figure 3: Include legends and labels on the figure

12) Pg 18, Line 424: Add figure reference (Fig 3B, C)

13) Table 3: What does NA mean in this context?

14) Typo: Fig 4 Legend: Delete “A, B)” before “Prenylation of the indicated naturally occurring”

15) Page 21, Line 495: Incorrect Figure reference. Should reference Fig 4C.

16) Line 504: Incorrect Figure reference. Should reference Fig 4C.

17) Line 411 pg 17: Cite previous studies

18) Table 4 and 5: The numbers are not displaying correctly.

Reviewer #2: The article titled "Functional classification and validation of yeast prenylation motifs using machine learning and genetic reporters" describes a new computational method to identify amino acid motifs in proteins that are prenylated. This method can also differentiate between prenylation followed by the canonical proteolysis and carboxy-methylation and the shunted, which only entails prenylation.

The manuscript if overall well-written and follows a logical structure.

The combination of the in-silico data and validation using yeast experiments is commendable.

It is clear that the new analysis tools are a substantial improvement of previously published methods.

Limitations are discussed.

Minor issues:

line 113 - thermotolerance does not describe the phenotype used here. "Growth defect at higher temperatures" would be more accurate.

line 307 - why do the different algorithms give such different results? Maybe it would be interesting to explain this.

line 446 - "some degree" can you be more specific?

In the Discussion the biological reasons why a protein might not be prenylated could be expanded (steric hindrance, protein-protein interactions etc.)

6. PLOS authors have the option to publish the peer review history of their article (what does this mean?). If published, this will include your full peer review and any attached files.

Reviewer #1: No

Reviewer #2: No

---

## [Author Response · Author response to Decision Letter 0]

21 May 2022

Editorial Concerns

(1) PLOS ONE's style requirements – manuscript and files fully adhere to requirements.

(2) Funding disclosure – details have been removed from the manuscript, and we request that the following be included on the online submission form:

“This work was supported by NIH funds to WKS and NK (NIH NIGMS GM132606, https://www.nih.gov/) and funds to WKS (NIH NIGMS R01GM117148, https://www.nih.gov/). The funders had no role in study design, data collection and analysis, decision to publish, or preparation of the manuscript.”

(3) Data access – There are no ethical or legal restrictions to data that is referenced in this study. All supporting data has been made available.

(4) Source images – all source images for preparation of figures are now reported as part of Supporting Information S1 Fig.

(5) References – all references are complete, correct, and now include a few new references that were suggested by Reviewer #1. This change altered the reference numbering from the original, but these changes were not tracked.

Reviewers' Criticisms

1. Is the manuscript technically sound, and do the data support the conclusions?

No concerns noted by either Reviewer.

2. Has the statistical analysis been performed appropriately and rigorously?

No concerns noted by either Reviewer.

3. Have the authors made all data underlying the findings in their manuscript fully available?

No concerns noted by either Reviewer.

4. Is the manuscript presented in an intelligible fashion and written in standard English?

No concerns noted by either Reviewer, but we made and tracked some minor edits that we believe will further improve readability.

Specific Criticisms Reviewer #1

1) The figures begin abruptly and it’s difficult to understand the first result. It would be helpful to include a cartoon figure at the beginning showing the canonical and shunted pathways and schematizing the workflow of the machine learning process.

Response: A new figure and legend have been included as suggested. The body of the manuscript now has new text that references the figure, and original figures have been renumbered to accommodate this change.

2) The rationale for use of the 19-sequence validation set was unclear. The authors should justify more clearly why only these 19 sequences were used.

Response: A clear rationale for use of the 19-sequence validation set is now provided.

3) For the uninitiated reader, it would be helpful to explain why the prenylated proteins migrate faster in the gel than the unmodified proteins, and what to expect in terms of mobility for cleaved vs. shunted proteins.

Response: We have now added comments to clarify that the effect of prenylation on protein migration by SDS-PAGE. Additionally, we have noted that there are few examples where cleavage has been observed to impact mobility (e.g., Ras GTPase).

4) Describe which features of Cxxx sequences are leading to the differences in the ESM-1b vs. Aaindex feature representation method results.

Response: A statement regarding the features of Cxxx sequencing leading to the differences in ESM-1b vs AAindex was added to the discussion.

5) Figures are in reverse order in the manuscript and some figures are mislabeled.

Response: Figure order and label issues have been addressed.

6) Figure 1a: Label how many data points are graphed. n=489+508?

Response: Suggested change was made.

7) Fig 1c: Label how many data points are graphed. n=136+140

Response: Suggested change was made.

8) Page 5, Line 115 should also cite prenylation effect on Ydj1-Hsp90 interaction: Flom GA, Lemieszek M, Fortunato EA, Johnson JL. Farnesylation of Ydj1 is required for in vivo interaction with Hsp90 client proteins. Mol Biol Cell. 2008 Dec;19(12):5249-58. doi: 10.1091/mbc.e08-04-0435. Epub 2008 Oct 1. PMID: 18829866; PMCID: PMC2592663.

Response: The indicated reference has been added along with minor text edits that place the reference in proper context.

9) Figure legend 2B: cleaved vs shunted figure legend and bar colors are reversed. Shunted should have probability = 0 of being cleaved.

Response: Suggested change was made.

10) Figure legend and bar colors in Fig 2A are also reversed.

Response: Suggested change was made.

11) Figure 3: Include legends and labels on the figure

Response: The figure and associated figure legend have been updated to provide better clarity to the reader.

12) Pg 18, Line 424: Add figure reference (Fig 3B, C)

Response: Text was modified as suggested.

13) Table 3: What does NA mean in this context?

Response: The Table 3 footnote was edited to better clarify as follows: “NA - not applicable; the non-prenylated status of the sequence precludes it from being cleaved; CaaX cleavage is prenyl-dependent.”

14) Typo: Fig 4 Legend: Delete “A, B)” before “Prenylation of the indicated naturally occurring”

Response: Typo corrected along with a few other minor typos were identified during preparation of the revised manuscript.

15) Page 21, Line 495: Incorrect Figure reference. Should reference Fig 4C.

Response: The Figure reference has been corrected as suggested, as well as another instance that we identified separately.

16) Line 504: Incorrect Figure reference. Should reference Fig 4C.

Response: The Figure reference has been corrected as suggested.

17) Line 411 pg 17: Cite previous studies

Response: The indicated reference has been added as suggested.

18) Table 4 and 5: The numbers are not displaying correctly

Response: The numbers are displaying oddly in the reviewer’s copy due to overlap with line numbering.

Specific Criticisms Reviewer #2

line 113 - thermotolerance does not describe the phenotype used here. "Growth defect at higher temperatures" would be more accurate.

Response: The suggested terminology has been incorporated.

line 307 - why do the different algorithms give such different results? Maybe it would be interesting to explain this.

Response: We have incorporated text in the Results section that address the variability in outputs by the different algorithms.

line 446 - "some degree" can you be more specific?

Response: We have incorporated more precise language to explain that the 12 sequences that were evaluated were either fully prenylated (100%) or mostly prenylated (75%).

In the Discussion the biological reasons why a protein might not be prenylated could be expanded (steric hindrance, protein-protein interactions etc.)

Response: We have moved and expanded the prior language on this topic to be more visible in the Discussion.

---

## [Decision Letter · Decision Letter 1]

6 Jun 2022

Functional classification and validation of yeast prenylation motifs using machine learning and genetic reporters

PONE-D-21-33646R1

Dear Dr. Schmidt,

We’re pleased to inform you that your manuscript has been judged scientifically suitable for publication and will be formally accepted for publication once it meets all outstanding technical requirements.

Kind regards,

Patrick Lajoie, PhD

Academic Editor

PLOS ONE

Additional Editor Comments (optional):

Reviewers' comments:

Reviewer's Responses to Questions

**Comments to the Author**

1. If the authors have adequately addressed your comments raised in a previous round of review and you feel that this manuscript is now acceptable for publication, you may indicate that here to bypass the “Comments to the Author” section, enter your conflict of interest statement in the “Confidential to Editor” section, and submit your "Accept" recommendation.

Reviewer #1: All comments have been addressed

Reviewer #2: All comments have been addressed

2. Is the manuscript technically sound, and do the data support the conclusions?

Reviewer #1: Yes

Reviewer #2: Yes

3. Has the statistical analysis been performed appropriately and rigorously? 

Reviewer #1: Yes

Reviewer #2: Yes

4. Have the authors made all data underlying the findings in their manuscript fully available?

Reviewer #1: Yes

Reviewer #2: Yes

5. Is the manuscript presented in an intelligible fashion and written in standard English?

Reviewer #1: Yes

Reviewer #2: Yes

6. Review Comments to the Author

Reviewer #1: The authors have addressed all the comments, and the manuscript is now suitable for publication

Reviewer #2: The revised version of the manuscript implemented all major chnages and additions suggested by the reviewers.

I think the manusciprt is no ready for publication.

7. PLOS authors have the option to publish the peer review history of their article (what does this mean?). If published, this will include your full peer review and any attached files.

Reviewer #1: No

Reviewer #2: No

---

## [Editor Report · Acceptance letter]

14 Jun 2022

PONE-D-21-33646R1 

Functional classification and validation of yeast prenylation motifs using machine learning and genetic reporters 

Dear Dr. Schmidt:

I'm pleased to inform you that your manuscript has been deemed suitable for publication in PLOS ONE. Congratulations! Your manuscript is now with our production department. 

Kind regards, 

on behalf of

Dr. Patrick Lajoie 

Academic Editor

PLOS ONE